# How Industry 4.0 and Sensors Can Leverage Product Design: Opportunities and Challenges

**DOI:** 10.3390/s23031165

**Published:** 2023-01-19

**Authors:** Albérico Travassos Rosário, Joana Carmo Dias

**Affiliations:** 1The Research Unit on Governance, Competitiveness and Public Policies (GOVCOPP), Universidade Europeia, 1200-649 Lisbon, Portugal; 2Centro de Investigação em Organizações, Mercados e Gestão Industrial (COMEGI), Universidade Lusíada, 1349-001 Lisbon, Portugal

**Keywords:** Industry 4.0, sensors, sensor technologies, product design

## Abstract

The fourth industrial revolution, also known as Industry 4.0, has led to an increased transition towards automation and reliance on data-driven innovations and strategies. The interconnected systems and processes have significantly increased operational efficiency, enhanced organizational capacity to monitor and control functions, reduced costs, and improved product quality. One significant way that companies have achieved these benefits is by integrating diverse sensor technologies within these innovations. Given the rapidly changing market conditions, Industry 4.0 requires new products and business models to ensure companies adjust to the current and future changes. These requirements call for the evolutions in product design processes to accommodate design features and principles applicable in the current dynamic business environment. Thus, it becomes imperative to understand how these innovations can leverage product design to maximize benefits and opportunities. This research paper employs a Systematic Literature Review with Bibliometric Analysis (SLBA) methodology to explore and synthesize data on how Industry 4.0 and sensors can leverage product design. The results show that various product design features create opportunities to be leveraged to guarantee the success of Industry 4.0 and sensor technologies. However, the research also identifies numerous challenges that undermine the ongoing transition towards intelligent factories and products.

## 1. Introduction

Companies are integrating technologies to design products that meet their customers’ growing needs and expectations. As a result, most firms are leveraging Industry 4.0 technologies associated with emerging intelligent factories and products that promise to transform the manufacturing process, thus impacting multiple market sectors [1]. Representatives from business, politics, and academia introduced the term “Industry 4.0” in 2011 through an initiative aiming to promote the idea as a technique to strengthen the German manufacturing sector. Bahrin et al. [2] define Industry 4.0 as the fourth industrial revolution involving digitization of the manufacturing sector through automation and data exchange in manufacturing technologies, including industrial Internet of Things (IoT), cyber–physical systems, cloud computing, and cognitive computing. In addition, sensors are critical components of Industry 4.0 since they connect various methods and devices, enabling multiple machines to communicate to track equipment and systems at each facility [3]. Consequently, incorporating sensors into Industry 4.0 technologies can enhance automation and sustainability and reduce costs through real-time output tracking and improved capability to monitor automated control systems.

Product design is critical in actualizing Industry 4.0 and developing sensor technology. This argument is evident in Tatipala et al.’s [4] research that stated that “without design, Industry 4.0 will fail” since the design is vital in accelerating the transformation of the manufacturing process. Product design under Industry 4.0 and sensors involves networking between different elements, such as machines and products [3]. For example, using cost-effective active sensors facilitates the data collection incorporated into Industry 4.0 to create intelligent, connected products, ensuring customer value creation. These processes can leverage product design principles that involve imagining, creating, and iterating products that address consumers’ specific needs within a particular market. Therefore, smart products manufactured through Industry 4.0 can be customized to match target markets’ needs and expectations, thus increasing competitiveness. However, Tatipala et al. [4] note that there is scarce research on product design in the context of Industry 4.0 despite promising benefits and opportunities. Therefore, this Systematic Literature Review with Bibliometric Analysis (SLBA) aims to identify the challenges and opportunities in integrating Industry 4.0 and sensors to leverage product design.

The structure of the paper is as follows. First, we explain the methodological approach used to respond to the object of study. The second section presents the bibliometric analysis carried out. The following section describes the theoretical perspectives resulting from the analysis carried out. Finally, we provide conclusions, implications, and future research directions.

## 2. Materials and Methods

In order to collect and synthesize the necessary data for this study, a Systematic Literature Review with Bibliometric Analysis (SLBA) was developed. According to Romanelli et al. [5], performing a bibliometric analysis allows researchers to assess the developments made in a given field, illustrating how the evidence connects to show the structure of the field. In this sense, the methodology was adopted due to its ability to unpack the evolutionary nuances in the fields of Industry 4.0, sensor technology, and product design, while also shedding light on emerging issues in these fields.

This process started with the definition of eligibility criteria to ensure that the results of the documents considered are accurate, objective, meaningful, and relevant to the study. Therefore, the researcher employed eligibility criteria for inclusion and exclusion.

The LRSB involves screening and selecting information sources to ensure the validity and accuracy of the data presented, in a process consisting of 3 phases and 6 steps [6,7,8,9] (Table 1).

This methodological approach focuses on bibliographical research in the online database for indexing scientific articles SCOPUS, one of the most important peer-reviewed databases in the academic world. The isolated use of Scopus is due to the fact that it is the main source of articles for academic journals/journals, covering about 19,500 titles from more than 5000 international publishers, including coverage of 16,500 peer-reviewed journals in a variety of scientific fields, thus providing a real view of the topics researched with scientific and/or academic relevance [7].

The methodological procedure started with the use of the keyword “Industry 4.0” in order to identify the appropriate sources in the Scopus directory. This initial search generated a total of 23,300 references. In order to reduce the high number of resulting sources, other search criteria were implemented based on the argument by Rosário and Dias [8] that only articles in journals and papers presented at conferences considered of “high quality” should be synthesized in a literature review, recommending that researchers adopt appropriate inclusion and exclusion criteria. Rosário and Dias [9] further explain that literary analysis improves readers’ understanding of the breadth and depth of existing literature. Therefore, to narrow the search to the most relevant literature, the keyword “sensors” was added, reducing it to 2870 documents, and later, a more specific keyword, “Product Design”, was added, restricting the document results to 26 scientific and/or academic documents (21 Conferences; 10 Articles; 2 Comments; and 2 Book Chapter) (Table 2).

## 3. Literature Analysis: Themes and Trends

The peer-reviewed documents were analyzed until October 2022. The year 2022 was the year with the highest number of peer-reviewed documents on the topic, with 16 publications. Figure 1 analyzes peer-reviewed publications published through October 2022.

We can say that in 2022, there has been an interest in research on Industry 4.0, sensors, and product design.

In Table 3, we analyze the Scimago Journal & Country Rank (SJR), the best quartile, and the H index by publication: *Computers In Industry* was the highest quality with 2430 (SJR), Q1, and an H index of 108. There is a total of seven publications in Q1, three publications in Q2, and four publications in Q3. Data from 12 publications are not available.

The thematic areas covered by the 26 scientific and/or academic documents were Computer Science (21); Engineering (21); Mathematics (5); Physics and Astronomy (5); Decision Sciences (4); Materials Science (4); Chemical Engineering (3); Business, Management and Accounting (2); Biochemistry, Genetics and Molecular Biology (1); Chemistry (1); Medicine (1); and Social Sciences (1). The most cited article was “Industrie 4.0 and smart manufacturing-a review of research issues and application examples” by Thoben et al. (2017) with 590 citations published in the *International Journal of Automation Technology* with 0.280 (SJR), the best quartile (Q3), and with an H index (20). The objective of this paper is to provide an overview of Industry 4.0 and smart manufacturing programs, analyze the application potential of CPS starting from product design through production and logistics up to maintenance and exploitation (e.g., recycling), and identify current and future research issues.

In Figure 2, we analyze the evolution of documents’ citations until October 2022. The number of citations shows a positive net growth with R2 of 79% for the year 2021 with 234 citations with a total of 811 citations.

The H index was used to verify the productivity and impact of published works, based on the largest number of articles included that had at least the same number of citations. Of the documents considered for the H index, eight were cited at least eight times.

In Appendix A, Table A1, citations of all scientific and/or academic documents until October 2022 are analyzed; nine documents were not cited in this period, making a total of 811 citations.

Figure 3 presents the bibliometric study to investigate and identify indicators of the dynamics and evolution of scientific information. The study of bibliometric results, using the scientific software VOSviewer, aims to identify the main research keywords in studies that are part of the research area of Industry 4.0, sensors, and product design. Here, we can see more clearly the most network nodes. The node size represents the occurrence of the keyword, i.e., the number of times the keyword occurs. The link between the nodes indicates the co-occurrence between the keywords, i.e., keywords that occur simultaneously or occur together, and its thickness reveals the occurrence of co-occurrences between the keywords, i.e., the number of times the keywords occur together or co-occur. The larger the node, the greater the occurrence of the keyword, and the thicker the link between the nodes, the greater the occurrence of co-occurrences between the keywords. Each color represents a thematic cluster, where the nodes and links in that cluster can be used to explain the topic coverage (nodes) of the theme (cluster) and the relationships (links) between the topics (nodes) that manifest under that theme (cluster).

The research was based on the analyzed articles about Industry 4.0, sensors, and product design. The associated keywords are presented in Figure 4, making clear the network of keywords that appear together/linked in each scientific article, thus allowing us to know the topics studied by the researchers and to identify future research trends.

The biggest nodes in this mapping are Lifecycle, 3D printers, and the Internet of Things. The results of the keyword development map from the Vosviewer are divided into three clusters. Cluster 1 is red with 17 keyword items, cluster 2 is green with 15 keyword items, and cluster 3 is blue with 6 keyword items, which can be seen in Figure 4 below. Cluster 1 is the largest cluster and refers to Lifecycle. These articles mainly focus on data mining, data analytics, advanced technology, blockchain, product lifecycles, and manufacturing process. Cluster 2 refers to 3D printers and focuses on issues such as technology transfer, change management, additive manufacturing, 3D printed mid, and production technology. Cluster 3, Internet of Things, involves cloud manufacturing, cloud manufacturing, cyber–physical systems, assembly, and computer-aided design. The three clusters are interconnected through Industry 4.0 and Product Design themes.

In Figure 5, a profusion of bibliographic couplings with a cited reference unit of analysis is presented.

## 4. Theoretical Perspectives

In today’s competitive business environment, companies face challenges such as customers’ demand for individualized products and short product lifecycles. In addition, there is an increase in the need to integrate software into hardware products to deliver higher customer value while simultaneously increasing operational efficiencies across the organization [10]. As a result, Industry 4.0 has become a famous development in recent years to address the need to interconnect machines, products, and people [11]. It raises collaboration productivity by facilitating linked systems, devices, and human resources to create quality, customized, high-value products. Sensors as critical components of Industry 4.0 have supported these goals by enabling data collection, analysis, and processing, thus supporting automation and real-time monitoring of systems and processes [12]. However, developing products under the Industry 4.0 context must adhere to new design principles and guidelines to create intelligent products. Therefore, this research section synthesizes data to demonstrate how Industry 4.0 and sensors can leverage product design to ensure that systems, processes, and products are developed to satisfy consumer needs and expectations (Table 4).

The industrial sector is crucial in every country’s economic growth since it is a key driver of economic growth and job creation. It involves manufacturing activities that transform raw materials into products, thus providing added value. With the increasing competition among manufacturing countries, developing and adopting advanced technologies to increase efficiencies and reduce costs have become a critical strategy [11]. As a result, the world has experienced an industrial revolution termed “Industry 4.0”, characterized by the broad application of technologies that significantly change established practices. For example, Industry 4.0 manufacturing plants have increasingly adopted automation and robots to increase operational efficiencies and reduce costs [2]. Data are the primary driver for this industrial revolution since this involves using advanced Information and Communication Technology (ICT) to connect multiple manufacturing machines, factories, units, raw material suppliers, customers, logistics enterprises, and energy suppliers [13]. The use and integration of ICT across all levels of the manufacturing process build a smart manufacturing network that benefits from automated, autonomic, and optimized manufacturing processes.

Industry 4.0, or the fourth industrial revolution, refers to the next phase of digitizing the manufacturing sector using technologies such as the Internet of Things (IoT), cyber–physical systems, and industrial Internet [38] and involves a combination of innovations, including software, sensors, processors, and communication technologies under the IoT and cyber–physical systems. These innovations are interconnected to facilitate information feeding into Industry 4.0, eventually adding value to the manufacturing processes [15]. The ultimate goal of Industry 4.0 is to create an open, smart manufacturing platform characterized by industrial-networked information applications [16]. Examples of technologies in Industry 4.0 include horizontal and vertical system integration, cybersecurity, the Internet of Things, the cloud, simulation, augmented reality, additive manufacturing (3D printing), big data analytics, and robots [2]. This networking is expected to allow companies easy and affordable access to modeling and analytical technologies that can be customized to meet each manufacturing company’s needs. Understanding Industry 4.0 technologies can help determine how product design can be leveraged across product development processes to ensure organizational productivity, performance, and customer satisfaction.

The horizontal and vertical system integration in Industry 4.0 reflects the evolution of cross-company, universal data-integration networks through automated value chains. The horizontal system integration involves Industry 4.0’s connected cyber–physical and enterprise systems networks [15]. It facilitates increased flexibility, automation, and operational efficiency throughout production. For example, machines and production units are interconnected across the production network, allowing them to communicate and autonomously respond to dynamic production requirements. On the contrary, vertical system integration involves connecting all business units and processes within the organization [16]. For instance, this aspect ensures a seamless data flow across all departments, from R&D, quality assurance, product management, IT, sales, and marketing. Therefore, horizontal and vertical system integration is the backbone of Industry 4.0 as it involves the interconnection of all units, processes, expertise, and systems within or cross-company.

The IoT refers to physical objects consisting of software, sensors, processing ability, and other technologies capable of connecting and exchanging data with other devices and systems over the Internet. In this case, IoT enhances a manufacturing company’s computing power by allowing devices to communicate and interact, thus decentralizing analytics and decision making and facilitating real-time responses [15]. Cybersecurity refers to protecting interconnected systems, devices, networks, servers, and data from malicious attacks. While Industry 4.0 provides opportunities for companies to improve operational efficiencies, productivity, performance, and competitiveness, the interconnectivity within the production networks and across the value chains increases vulnerability to cyber security threats [17]. Therefore, cybersecurity technologies are critical success components of Industry 4.0 as they guarantee the safety and security of the systems and processes. Cloud technologies provide computing services such as servers, storage, databases, networking, software, analytics, and intelligence over the Internet, known as “the cloud”, to support data-driven production processes [17]. These cloud computing technologies have increasingly become critical in Industry 4.0 as they allow data sharing across sites and company confines.

Big data and analytics technologies allow the collection and comprehensive data analysis from multiple sources and customers. As a result, these innovations support real-time decision making, optimize production quality, and help reduce energy consumption and equipment maintenance [39]. On the other hand, simulations are used to leverage real-time data in a virtual framework to mirror a physical world, including machines, products, and humans. These technologies allow developers to test and optimize outcomes before introducing them to the actual application [17]. For example, simulations can adjust machine settings until the operator achieves the required performance levels.

Similarly, additive manufacturing uses computer-aided design (CAD) or 3D object scanners to create objects with precise geometric shapes. It is widely used in Industry 4.0 to produce small batches of customized products that provide construction benefits such as complex, lightweight designs [19]. Other technologies, such as augmented-reality-based systems, send repair instructions or select parts in a warehouse using mobile devices. At the same time, robots are expected to increasingly become more autonomous, flexible, and cooperative to work safely alongside humans [20]. Although each of these Industry 4.0 technologies has distinct features and functionalities, interlinking them to form a complex interconnected network to enhance collaboration and operational efficiency.

### 4.1. Sensors under Industry 4.0

Various industries use different types of sensors for varying applications in routine and commercial purposes. Sensors link multiple devices and systems and allow machines to communicate and track equipment and systems at each facility. Javaid et al. [3] (p. 2) define a sensor as a “device that detects the input stimulus, which may be any quantity, property, or condition from the physical environment, and responds to a measurable digital signal.” Examples of input stimulus include environmental conditions such as temperature, pressure, moisture, heat, force, or light, while the response output has a frequency, resistance, capacitance, current, and voltage. As an independent system, sensors can process onboard by assessing the atmosphere conditions and changing operations accordingly [21]. The sensor technology in Industry 4.0 can collect and analyze large quantities of data rapidly and accurately to stimulate appropriate actions. This increased capacity eliminates issues such as human error, reduces the need to monitor systems, and enhances quality and production. Therefore, sensors are critical innovations in Industry 4.0.

Industry 4.0 is characterized by integrated and smart networks that facilitate intelligent production. Sensors have multiple features and capabilities that make them essential for the success of Industry 4.0. For instance, they can capture and analyze data for appropriate decision making and promote automation across production lines by enabling self-optimization [21]. Other than the overall process automation, sensors have additional features such as predictive maintenance and asset monitoring or conditioning [22]. The current manufacturing sector is rapidly transitioning to automated technologies to improve production processes and product quality [23]. In this case, leveraging software intelligence will require advanced sensor technologies to optimize manufacturing activities, monitor processes, and increase efficiencies. According to Lo et al. [24], sensors’ capabilities and capacities are widely applied in pharmaceutical and chemical plants and industrial robots, where sensor technologies are used for multiple applications, including process control through flow calculation and temperature sensors. In addition, smart sensors are used to evaluate dynamic circumstances and environmental changes throughout the manufacturing processes.

Industry 4.0 is primarily data-driven, with technologies used in data collection, analysis, and interpretation playing the most critical roles. Sensors contribute to these practical applications by performing multiple vital functions, such as providing raw data that reveal broader inefficiencies affecting machine performance [24]. For example, smart sensing technologies collect data on temperature range, flow, pressure response, and measuring fluids. In addition, intelligent sensors are used in real-time data gathering, remote surveillance, and preventive maintenance, thus enhancing equipment performance. Javaid et al. [3] indicate that sensor technologies are used in manufacturing to perform multiple functions, including providing product details, ensuring precise positioning by giving feedback on motor movement, and issuing alerts on equipment conditions. In addition, sensors can be used to determine extrinsic and intrinsic characteristics of objects, including location, distance, proximity, temperature, and color [23]. Therefore, sensor technologies integrated into automation systems in Industry 4.0 provide a critical way of collecting data on procedures and production activities.

### 4.2. Product Design

The product design process involves creating a business innovation based on a market opportunity, a defined problem related to people’s needs, technological possibilities, and business feasibility. It can be described as the process in which the designer establishes design decisions using various product data and transforms functional requirements into a specific implementation structure. Chen et al. [25] describe product design as a complex iterative process that begins with creating a product’s principle scheme design, followed by the overall design and the final detailed scheme design. As a result of this complexity, the product design process is often broken down into various tasks and processes and characterized by a clear division of labor where each activity is assigned to specific skilled staff and departments [26]. Despite the allocation of functions throughout the design process, the involved experts collaboratively work to develop innovations that meet specific customer needs.

The product design process can include multiple phases. This study focuses on three main stages: requirement analysis, conceptual design, and detailed design.

#### 4.2.1. Requirement Analysis

During this phase, the designers and their teams analyze key customer preferences and translate them into useful product features and attributes. Appropriate methods are used to obtain customer requirement information, which is then screened for relevance and impact on product design [27]. For every manufacturing firm operating in the current competitive business environment, the primary motivation is to meet customer requirements by designing and producing products that directly address their problems. This necessity makes this initial stage of the product design process critical since it ensures that all product features align with customer needs and expectations [38]. As a result, Industry 4.0 technologies such as big data and the Internet of Things are used to collect and analyze data on customer preferences and expectations, and insights are integrated into product design decision making.

#### 4.2.2. Conceptual Design

This stage involves a series of iterative and complex engineering processes where relevant knowledge is combined to establish a functional structure and search for a proper working principle. This stage pursues the correct combination mechanism, defines the basic solution course, and generates the design scheme [40]. Design concept generation is a critical determinant of the success of new product development. For instance, companies must develop products that meet their target customers’ diverse, individualized needs while maintaining low production costs and product development cycles [26]. The conceptual design phase can solve these issues by ensuring the efficient use of product data to generate a concept that considers consumer preferences and needs, resource availability, and profitability. Traditionally, designers have relied on their knowledge and experience in conceptual product design. However, Industry 4.0 technologies have shifted this by increasing designers’ access to data and innovations that lead to improved product design quality.

#### 4.2.3. Detailed Design

This phase models the product development process using data to create product solutions based on requirements and the structure built in the first and second stages. For instance, the designer determines product features such as appearance, configurations, and parameter data [40]. In this case, the designers use the predefined product concept to complete a product’s essential aspects, including desired performance levels and design information. Developing sensors and data storage technologies in Industry 4.0 avails large volumes and types of products that can aid the designing process [29]. As a result, data mining and database technologies are critical in developing and applying data-driven modeling methods in product design [30]. The data integrated into the design modeling process provide the rationale for the requirements considered in creating the detailed design. For instance, the data can be used to prove why the adopted design is the most appropriate solution to the predefined problem serving as the foundation for the product design process.

### 4.3. Opportunities in Leveraging Product Design in Industry 4.0 and Sensors

Industrial revolutions are associated with multiple benefits, including increased production performance through new technologies, reduced costs, and increased ability to develop new, affordable products that meet current market needs. As a result, creating a compatible product design and development process (PDDP) becomes necessary for companies to exploit these benefits [31]. Therefore, the success of Industry 4.0 and sensor technology depends on manufacturing companies’ capacity to develop new products and business models adjusted to fit the rapidly changing market conditions [29]. Smart factories can support the production of customized products, while smart products have a higher potential to satisfy customer needs through information exchange and adaptiveness. However, research shows these benefits can only be achieved if the products are designed appropriately. Therefore, as the manufacturing sector transitions towards smart factories and smart products, the interdependence between Industry 4.0 technologies and innovations, including sensors and product design engineering, becomes increasingly deeper [32]. This section explores design features and opportunities that can be leveraged in the context of Industry 4.0 and sensors to guarantee continuous development and improvements.

#### 4.3.1. Design for Empowered Users/Customers

Customer empowerment in product design requires customer involvement throughout the process. It can occur during the final product configuration definition, where the designer creates building blocks instead of finalized products, or during the production process, where the designer recognizes customers’ capability to produce the final products [41]. Designing building blocks require an enormous understanding of the problem in which the solution is to be established. Suppose the customers are involved at this stage of product design. They must understand the combination of various design elements, shape, texture, color, negative/white space, and value [34,38]. However, their involvement improves the probability of high adoption rates since it means the final product will be easy to use by target customers since they were actively involved. Industry 4.0 technologies applicable in this designing phase include the cloud and augmented reality (AR). Marino et al. [35] explain that AR is a critical enabling technology in Industry 4.0 that integrates virtual information into a user’s real-world perception by combining vision and sensor-based tools. In addition, the authors indicate that AR can be used to assess actual products to identify design discrepancies between the planned features and products developed. In the context of customers as configurators, AR can be used to enable customers to assess the design elements to ensure the right combination before settling on a solution.

The second customer empowerment phase during the production process explores the possibility that customers produce the final products. In this case, the customers are encouraged to own the manufacturing process. Given the increased competition and customer awareness in the current business environment, companies increasingly involve their target customers in production [38]. Industry 4.0, as a data-driven industrial revolution, has seen an increase in customers’ access to critical information that influences their perception of a product or the company itself. Therefore, leveraging this design feature can help create competitive products and improve a company’s competitive position in the market [41]. The empowerment is achieved through multiple techniques, including customer education, availing user-friendly processes and tools, and ensuring a flexible, viable producing capacity. The Industry 4.0 technologies considered in this design phase include cloud and additive manufacturing (AM). The cloud facilitates data storage and sharing, thus contributing to empowerment through access to relevant knowledge and allowing customer contributions [17]. AM increases manufacturing operations’ flexibility and efficiency by using computer-aided design (CAD) software or 3D object scanners to deposit material layer upon layer in precise geometric shapes [20,36]. The CAD software can be used to demonstrate customer information and digitally define objects, thus illustrating how the final product will look and creating an opportunity for feedback and further improvements.

#### 4.3.2. Design for Cyber Security

Industry 4.0 is based on cyber–physical systems, where data, vertical and horizontal data integration and exchange, and data analytics play a central role. While these data innovations create opportunities to enhance efficiency, performance, productivity, and product quality, they also bring cyber-environment challenges, such as hackers and viruses, to the physical world [37]. In this case, the design engineer must guarantee data security by strengthening safety, security, privacy, and knowledge protection controls and measures [41]. Design for cyber security requires that the product design processes prioritize security as the first principle in design and across the value chain. Therefore, the design team must have IT experts who understand potential vulnerabilities associated with adding software and IoT into design solutions and possible countermeasures [38]. Systems designed for cyber security have higher capacities to meet individual needs since they can perform functions linked to sensing and responding, autonomy, and configuration.

The interconnection in Industry 4.0 facilitates the optimization of the increasing data density and the fusion of information and operational technologies. However, it increases vulnerability to cyber threats making cyber security a core issue in Industry 4.0 [41]. In addition, sensors play a critical role in the initial stages of Industry 4.0, where they are attached to industrial assets to establish digital records by collecting data through imitation of human feelings and thoughts [33]. Consequently, research shows that sensor-based platforms and applications are highly vulnerable to cyberattacks [19]. Considering these circumstances, leveraging the product design process can help mitigate the cyber security issues in Industry 4.0 and sensor technologies by ensuring that the proposed and developed designs prioritize safety, security, privacy, and knowledge protection.

#### 4.3.3. Design for Data Analytics

Data modeling and data analytics are crucial in the product design process. According to Cattaneo et al. [14], data analytics involves data mining and statistical models needed at the beginning to clean data and validate the rules. The IoT-based manufacturing sector in Industry 4.0 has led to the generation of a tremendous amount of data that requires analysis through multiple methods, including artificial intelligence, machine learning, and data mining [42]. Data collected about products, markets, or customer needs can provide design knowledge, creating opportunities to improve product competitiveness and production efficiency. In addition, data-driven product design involves using data modeling and analysis to uncover hidden patterns and relevance to enhance product and system schemes [14]. Product data are generated throughout the product lifecycle and through the interactions between products, humans, and the environment. Industry 4.0 innovations such as cyber–physical systems, digital Internet resources, and scientific experiments are critical product data sources. Other technologies that can be leveraged for data access include simulation and horizontal and vertical systems integration [18]. Therefore, data analytics leads to better product design decisions and improves customer satisfaction and organizational competitiveness since the designers better understand the target users, thus developing individualized tools and resources. This design feature can be integrated into Industry 4.0 and sensor technologies to ensure that smart factories align their goals and practices toward addressing customers’ specific needs.

#### 4.3.4. Design for Changeability

High dynamics and increased individualization characterize the current manufacturing sector. As a result, it has become critical for companies to have the ability to adjust their production systems to future needs and conditions quickly. Therefore, design for changeability features is concerned with designing systems and products with built-in robustness against slight use variations and potential future changes [27]. For example, intelligent factories and products require flexible updating and upgrading, the ability to accommodate new technologies, and adaptability to diverse user experiences. In this case, the designers must understand factors that may drive future changes in the product and then determine changeable solution architectures [41]. For example, dynamic marketplaces and technical evolution can lead to the need for changes. Design for changeability is described from four primary perspectives; robustness, flexibility, agility, and adaptability. Robustness refers to a system’s insensitiveness towards changes, while flexibility focuses on the capacity to easily accommodate changes. Agility refers to the ability to change rapidly, while adaptability refers to the ability to adjust to changing circumstances [43]. As the manufacturing sector transitions towards smart factories and smart products under Industry 4.0, design for changeability has become a critical feature as it facilitates the creation of systems and products that can quickly and rapidly adjust to the changes. Therefore, the design for changeability principles should be leveraged when designing Industry 4.0 and sensor technology architectures.

### 4.4. Challenges in Leveraging Product Design

While multiple product design features can be leveraged in Industry 4.0 and sensor technologies, various challenges hinder their optimization. For instance, although there has been abundant research on Industry 4.0 and sensors, there is limited research on how they can leverage product design. It, therefore, becomes challenging to implement product design features and principles in smart factories and products due to a lack of adequate information and critical insights on how that can be achieved. Other challenges identified in the research include infrastructure constraints, lack of technological competencies, and legal issues relating to privacy and security.

#### 4.4.1. Infrastructure Constraints

Advanced technologies used in Industry 4.0 and sensors are expensive, limiting some companies, especially SMEs, from leveraging product design features and associated opportunities. Maximizing benefits linked to the correlation between product design and Industry 4.0 innovations, including sensors, require access to advanced ICT infrastructure to facilitate the establishment of cyber–physical systems and IoT-based applications and systems [29]. Despite the rapid technological advancements, manufacturing plants still struggle with immature IT to support the integration of Industry 4.0 technologies and facilitate the smooth transition towards intelligent factories and products. Therefore, infrastructural factors remain a significant barrier to leveraging product design in Industry 4.0 and sensor technologies.

#### 4.4.2. Technological Competencies

Lack of skills and knowledge on Industry 4.0 innovations and how to effectively integrate them into organizational systems and processes is a significant challenge. The authors in [20] indicate that most traditional manufacturing companies struggle with a skills gap due to the high number of aging populations in their workforce. Older employees have limited tech skills and knowledge, which limits their ability to embrace and adopt advanced technologies. As a result, these companies must develop strategies and programs to increase employees’ skills and raise awareness on topics such as deep simulation and coding experience [38]. These training and awareness programs can lead to slow adoption rates since the employees may take time before they have adequate skills and knowledge to adopt the advanced technologies. Moreover, the process can also be influenced by employees’ readiness to adopt the technologies [27]. For example, if the older workforce has negative attitudes towards the new technologies, the adoption rate or participation in the training programs may be lower. Alternatively, these companies can hire new employees with the necessary tech skills and knowledge. While these practices can increase an organization’s competitiveness, they may be expensive, thus leading to organizational reluctance to adopt advanced technologies.

#### 4.4.3. Legal Issues

Despite the massive research and awareness of cyber security and its impact on Industry 4.0, security and privacy concerns remain significant challenges. As a result, most governments have implemented regulations on data protection and IT security, liability, and intellectual property laws. Companies must meet compliance standards and ensure their practices are within the defined protocols to avoid legal issues, such as lawsuits. In addition, companies are expected to adhere to organizational or industry standards, codes, principles of good governance, and ethical and social norms [28]. However, compliance can be challenging due to laws varying across territories. For example, most companies function in other countries worldwide with emerging technologies. Differences in data protection and IT security laws and policies can significantly undermine these companies’ compliance efforts.

## 5. Conclusions

This research identifies multiple design features that can be leveraged in Industry 4.0 and sensor technology to facilitate smooth transition and development. Firstly, design for empowered users/customers advocates for active engagement of target customers throughout the designing and product production processes. This opportunity is critical in Industry 4.0 since it ensures product development addresses customer needs and expectations directly. Secondly, design for cyber security can be leveraged in Industry 4.0 and sensors to reduce vulnerability to cyber-security threats. Cyber security has become a significant issue due to the increased interconnection in Industry 4.0 networks. Thus, leveraging this design feature can help mitigate the problem. Design for data analytics reinforces the significance of adopting data-driven designs and processes to ensure that organizational practices and products meet market needs. Finally, design for changeability ensures that companies adopt production systems and processes that can easily and quickly adjust to future changes and variations. However, various challenges often hinder leveraging these opportunities and features, including infrastructural constraints, inadequate technological competencies, and legal issues relating to security and privacy concerns. As companies continue to integrate Industry 4.0 technologies, they must develop practical solutions addressing these issues to ensure they benefit from the opportunities presented by emerging, advanced innovations.

This article has some limitations, mainly in the selection and use of databases and in the keywords chosen. Although Scopus is the largest database, there are publications indexed in other databases that might be extremely important. Furthermore, the theme of this article focuses on how Industry 4.0 and sensors can leverage product design so other variables or other aspects are not explained in detail. As for the keywords used in the research, we consider that the term Industry 4.0 can be reductive in the search. For future research, we expect to use other databases such as EBSCO and ISI Web of Science to be able to see a map of the development of search trends, and also to use other keywords related to the term Industry 4.0.

## Figures and Tables

**Figure 1 sensors-23-01165-f001:**
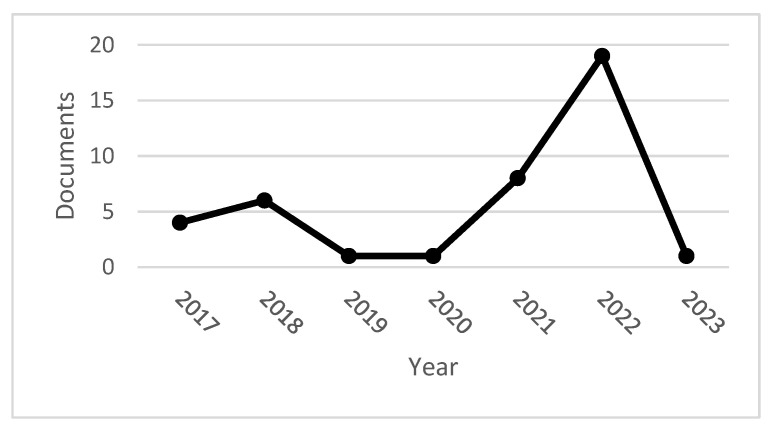
Documents by year. Source: own elaboration.

**Figure 2 sensors-23-01165-f002:**
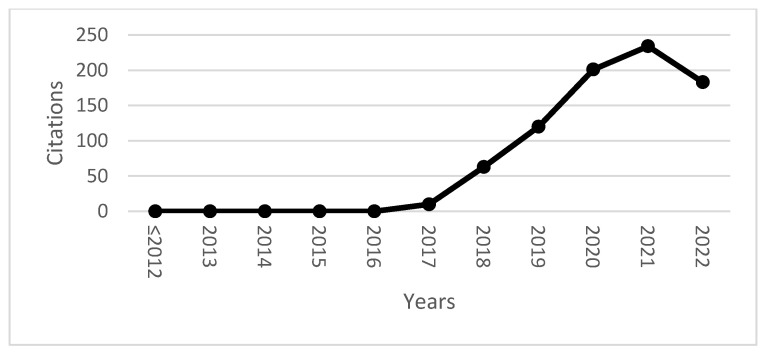
Evolution of citations between ≤2012 and October 2022. Source: own elaboration.

**Figure 3 sensors-23-01165-f003:**
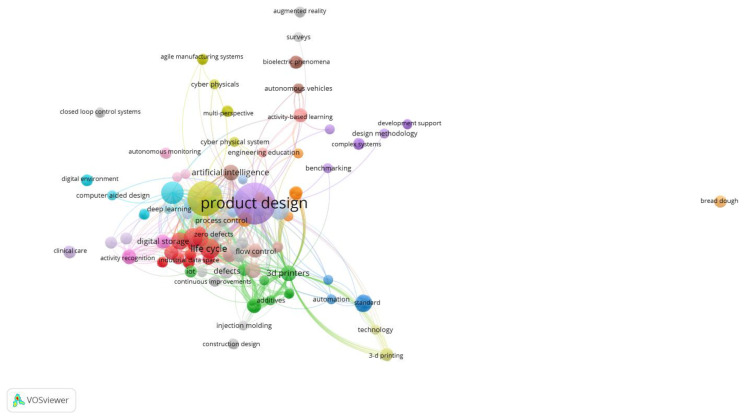
Network of all keywords.

**Figure 4 sensors-23-01165-f004:**
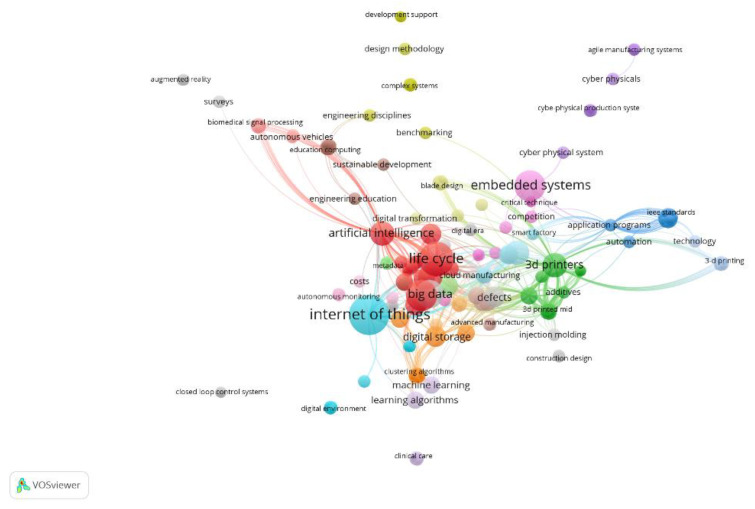
Network of Linked Keywords.

**Figure 5 sensors-23-01165-f005:**
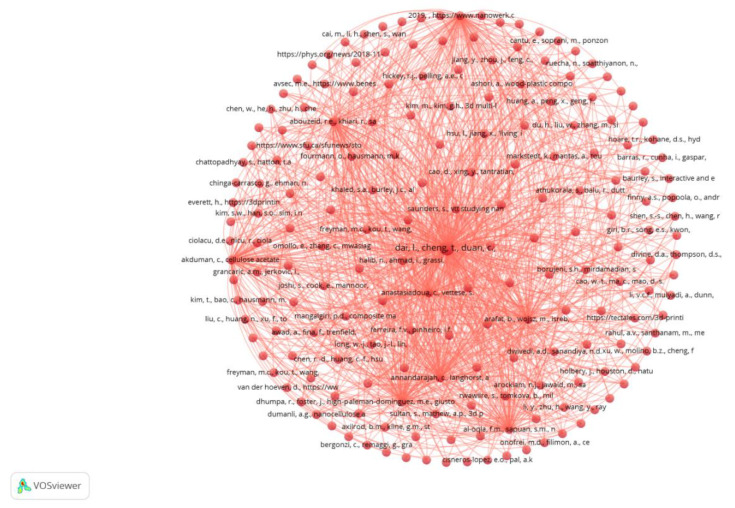
Networks bibliographic coupling.

**Table 1 sensors-23-01165-t001:** Process of systematic literature review with bibliometric analysis (SLBA).

Fase	Step	Description
Exploration	Step 1	formulating the research problem
	Step 2	searching for appropriate literature
	Step 3	critical appraisal of the selected studies
	Step 4	data synthesis from individual sources
Interpretation	Step 5	reporting findings and recommendations
Communication	Step 6	Presentation of the LRSB report

Source: [7].

**Table 2 sensors-23-01165-t002:** Screening Methodology.

Database Scopus	Screening	Publications
Meta-search	keyword: Industry 4.0	23,300
Inclusion Criteria	keyword: Industry 4.0, sensors scientific and/or academic documents high-quality publications English language	2870
	keyword: Industry 4.0, sensors, Product Design scientific and/or academic documents high-quality publications English language	26
Screening	Published until October 2022	

Source: own elaboration.

**Table 3 sensors-23-01165-t003:** Scimago Journal & Country Rank impact factor.

Title	SJR	Best Quartile	H Index
*Computers In Industry*	2430	Q1	108
*Applied Soft Computing Journal*	1960	Q1	156
*Materials Chemistry Frontiers*	1590	Q1	58
*Quality And Reliability Engineering International*	1020	Q1	67
*Journal Of Mechanical Design Transactions Of The ASME*	1010	Q1	126
*International Journal Of Advanced Manufacturing Technology*	0.920	Q1	134
*Sensors*	0.800	Q1	196
*Future Internet*	0.790	Q2	38
*Procedia CIRP*	0.640	- *	78
*Procedia Computer Science*	0.570	- *	92
*Production Engineering*	0.520	Q2	35
*International Journal On Interactive Design And Manufacturing*	0.490	Q2	28
*International Journal Of Automation Technology*	0.280	Q3	20
*IFIP Advances In Information And Communication Technology*	0.250	Q3	56
*ACM International Conference Proceeding Series*	0.230	- *	128
*Ceur Workshop Proceedings*	0.230	- *	57
*Sensors And Materials*	0.220	Q3	29
*Smart Innovation Systems And Technologies*	0.220	Q3	27
*2019 IEEE International Workshop On Metrology For Industry 4.0 And Iot Metroind 4.0 And Iot 2019 Proceedings*	0	- *	8
*Advances In Transdisciplinary Engineering*	0	- *	12
*Closer 2018 Proceedings Of The 8th International Conference On Cloud Computing And Services Science*	- *	- *	- *
*2021 14th International Congress Molded Interconnect Devices Mid 2021 Proceedings*	- *	- *	- *
*2021 IEEE International Workshop On Metrology For Industry 4.0 And Iot Metroind 4.0 And Iot 2021 Proceedings*	- *	- *	- *
*2022 16th European Conference On Antennas And Propagation Eucap 2022*	- *	- *	- *
*Eai Springer Innovations In Communication And Computing*	- *	- *	- *
*VDI Berichte*	- *	- *	- *

Note: * data not available. Source: own elaboration.

**Table 4 sensors-23-01165-t004:** Articles and scientific documents from SCOPUS database.

Title	Authors	Journal	Object of Study and Main Conclusions
Presents the technology, protocols, and new innovations in industrial internet of things (IIoT)	[13]	*EAI/Springer Innovations in Communication and Computing*	Smart devices are changing people’s daily life in the world, in which significant trend has already been extended to the industry sector. In the upcoming Industry 4.0, the connected smart devices all around the world via the Internet provide secure, real-time, and reliable services of sensing, communicating, and computing, making smart factories into realization.
Clarifying Data Analytics Concepts for Industrial Engineering	[14]	*(No source information available)*	The paper provides an overview of the data analysis techniques that could be used to extract knowledge from data along the manufacturing process.
The digital twin in Industry 4.0: A wide-angle perspective	[15]	*Quality and Reliability Engineering International*	This paper is about surrogate models, also called digital twins, that provide an important complementary capacity to physical assets. Digital twins capture past, present, and predicted behavior of physical assets.
Bayesian inference for mining semiconductor manufacturing big data for yield enhancement and smart production to empower industry 4.0	[16]	*Applied Soft Computing Journal*	This study aims to develop a framework based on Bayesian inference and Gibbs sampling to investigate the intricate semiconductor manufacturing data for fault detection to empower intelligent manufacturing.
Smart connected digital factories unleashing the power of industry 4.0 and the industrial internet	[17]	*CLOSER 2018—Proceedings of the 8thInternational Conference on Cloud Computing and Services Science*	This paper focuses on Industry 4.0 technologies and how they support the emergence of highly connected, knowledge-enabled factories, referred to as Smart Manufacturing Networks.
Development of a Predictive Maintenance 4.0 Platform: Enhancing Product Design and Manufacturing	[18]	*Smart Innovation, Systems and Technologies*	The purpose of this study is to investigate and explore the potential of predictive maintenance and its relation to Industry 4.0, and product/process re-engineering through product lifecycle management (PLM), hence leading to Predictive Maintenance 4.0.
Evaluation of different additive manufacturing technologies for MIDs in the context of smart sensor systems for retrofit applications	[19]	*14th International Congress: Molded Interconnect Devices, MID 2021- Proceedings*	In the context of this paper, three additive technologies are evaluated with respect to their applicability against the background of different retrofitting applications. A focus lies on the creation of 3D-shaped circuit carriers.
An industry 4.0 framework for tooling production using metal additive manufacturing-based first-time-right smart manufacturing system	[20]	*Procedia CIRP*	This paper presents a concept for an integrated process chain for tooling production based on metal additive manufacturing.
A Sensor Data Fusion-Based Locating Method for Reverse Engineering Scanning Systems	[21]	*IEEE International Workshop on Metrology for Industry 4.0 and IoT, MetroInd 4.0 and IoT 2019*	The present paper faces the locating problem of a handling device for reverse engineering scanning systems. It proposes a locating method by using sensor data fusion based on Kalman filter, implemented in a Matlab environment by using low-cost equipment.
Predictive Maintenance in Industry 4.0	[22]	*ACM International Conference Proceeding Series*	This paper looks at how to support predictive maintenance in the context of Industry 4.0.
Unsupervised learning for product use activity recognition: An exploratory study of a “chatty device”	[23]	*Sensors*	This paper proposes a model that enables new forms of agile engineering product development via “chatty” products. Products relay their “experiences” from the consumer world back to designers and product engineers through the mediation provided by embedded sensors, IoT, and data-driven design tools.
Design of Injection Molding of Side Mirror Cover	[24]	*Sensors and Materials*	The purpose of this paper is to develop a design for the injection molding of the product, which is applied in the concept of Industry 4.0 that aims to have intelligent processes.
Utilizing cyber physical system to achieve intelligent product design: A case study of transformer	[25]	*Advances in Transdisciplinary Engineering*	This study utilizes the framework of CPS to achieve intelligent product design.
Towards Smart Assembly Based Design	[26]	*Lecture Notes in Mechanical Engineering*	This paper proposes a new framework of data-driven smart assembly design to keep pace with the industrial and Information Technology (IT) revolution.
The implementation of Industry 4.0 in manufacturing: from lean manufacturing to product design	[11]	*International Journal of Advanced Manufacturing Technology*	With interconnection through Industry 4.0, upgraded legacy machinery can provide more in-depth and detailed process information, which, as well as enabling process improvements, can inform the product design to achieve higher production efficiency.
Electrospindle 4.0: Towards Zero Defect Manufacturing of Spindles	[27]	*CEUR Workshop Proceedings*	In this paper, the authors discuss the goals of the Electrospindle 4.0 project, which aims at applying Zero Defect Manufacturing principles to the production of spindles.
Preliminary Design of a Double Ridge Waveguide Device for Monitoring the Complex Permittivity of Carasau Bread Doughs	[28]	*16th European Conference on Antennas and Propagation, EuCAP 2022*	This work deals with the preliminary design of a double ridge waveguide device to perform indirect measurements of the complex permittivity of a traditional food product from Sardinia (Italy), i.e., the Carasau bread, in the case of a small bakery industry.
Barriers for industrial sensor integration design-an exploratory interview study	[29]	*Journal of Mechanical Design, Transactions of the ASME*	The aim of this paper is to explore potential challenges within different contexts and suggest possible directions for research within the field of sensor integration design.
Modeling Fused Filament Fabrication using Artificial Neural Networks	[30]	*Production Engineering*	This study uses a trained artificial neural network (ANN) model as a digital shadow to predict the force within the nozzle of an FFF printer using filament speed and nozzle temperatures as input data.
Lean thinking in the digital Era	[31]	*IFIP Advances in Information and Communication Technology*	This paper describes the current state of the art in order to understand how lean thinking should be implemented in the context of the smart factory.
A vest for treating jaundice in low-resource settings	[32]	*IEEE International Workshop on Metrology for Industry 4.0 and IoT, MetroInd 4.0 and IoT 2021 -Proceedings*	This paper aims to address the issues and causes of insufficient NJ phototherapy on a global scale, presenting the design, test, and development of a first prototype of a vest, with embedded fiber optics and sensors for autonomous phototherapy treatment of newborn jaundice in LRSs.
Emotion recognition for semi-autonomous vehicles framework	[33]	*International Journal on Interactive Design and Manufacturing*	This article proposes a novel approach for emotion recognition that not only depends on images of the face, as in the previous literature, but also on the physiological data
S4 Features and Artificial Intelligence for Designing a Robot against COVID-19—Robocov	[34]	*Future Internet*	This paper shows how a reconfigurable robot can be designed under the S3 concept and integrate AI methodologies.
An Augmented Reality inspection tool to support workers in Industry 4.0 environments	[35]	*Computers in Industry*	In this paper, an innovative AR tool has been proposed to assist workers at the workplace during inspection activities of industrial products.
3D printed cellulose based product applications	[36]	*Materials Chemistry Frontiers*	This review highlights the many promising and diverse functions and applications of sustainable 3D-printed cellulose-based products.
NETra model at Rajarambapu Institute of Technology (RIT):Transform engineering campus into product innovation centre -Journey so far	[37]	*Procedia Computer Science*	With rapid advancements in Industry and the pace of the Industry 4.0 revolution, Engineering Education must enhance project-based learning (Project BL) methodology to product-based learning (Product BL). The motivation of this study is to transform engineering campuses into Product Innovation Centers.
Foundry 4.0: Smart casting process control and real time quality prediction: The digitalization of foundry plays a key role in competitiveness introducing new integrated platform to control the process and predict in real-time the quality and the cost of the casting	[12]	*VDI Berichte*	The FP7- MUSIC project is giving a new age to the traditional multi-stages production processes such as High-Pressure Die Casting (HPDC). The use of Sensors and totally integrated systems, as well as the data mining and cognitive model, are the key ingredient of the MUSIC project to be a reference in the Industry 4.0 context.
“Industrie 4.0” and smart manufacturing—a review of research issues and application examples	[10]	*International Journal of Automation Technology*	The objective of this paper is to provide an overview of Industry 4.0 and smart manufacturing programs, analyze the application potential of CPS starting from product design through production and logistics up to maintenance and exploitation (e.g., recycling), and identify current and future research issues.

Source: own elaboration.

## Data Availability

Not applicable.

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
