# Peer review of "How Industry 4.0 and Sensors Can Leverage Product Design: Opportunities and Challenges"

_sensors, 2023, doi:10.3390/s23031165_

Round 1

Reviewer 1 Report

This paper aims to understand the role of Industry 4.0 in product design. A review of 35 papers is presented. The review is presented as follows:

Abstract:

horizontal and vertical system integration is not a technology

I suggest improving the presentation of the results. The part “

 The results show” is the most important in the abstract and will increase the chances of capturing the readers' attention. The introductory parts with general concepts about industry 4.0 can be reduced.

Finally, I suggest restructuring the part “While numerous research has been conducted to show the role (…) business environment”, not starting with a sentence that few studies addressed the topic but clearly stating the authors' research objective.

Introduction

The second part starts with a confusing sentence. In the initial parts, it seems that the authors will address sensors for product design, but the sentence points to the design of sensors

I suggest concluding the Introduction with the paper structure.

Section 2

“bibliometric literature” does not sound appropriate. It could be scientific literature or a bibliometric analysis (which is a different approach– it is possible to conduct one without the other)

Please explain the meaning of the letters in LRSB and provide a reference. Table 1 also needs references if the steps are traditional in that approach.

Page 2 line 84 requires a reference. There are other important databases like web of science, among others. Moreover, the sentence “However, we assume…” should be removed and included in the limitations section. In my opinion, it is valid to select Scopus, and many authors use it, so it is better just to explain the reason for selecting this important database (not the most important) with a supporting reference.

The paragraph “The procedure started with using the keyword” is not clear. It seems that the authors screened the titles, abstracts, and keywords of 23,300 references, which seems highly improbable. Not even the 2,870 records seem feasible to screen. The authors seem to be explaining the keyword testing and experiments before selecting the better focus of the analysis, not the screening process. I suggest including a figure of the process or improving the presentation because it is not clear how the 35 papers appear (seems to be the result of TITLE-ABS-KEY ( "Industry 4.0"  +  AND sensors  AND +  "Product Design" but it is not stated clearly).

To increase the confusion, the authors state that “only articles in journals considered as 95 "high quality" should be synthesized in a literature review” but most of the papers are presented at conferences.

The authors found related work? Reviews about product design in Industry 4.0? The authors state that studies are limited but examples of existing studies and how this paper is difference could be presented.

Table 2 needs corrections. Inclusion criteria are not related to keywords. Restricting to the type of publication or the data are criteria, not the keywords.

After madding some tests (e.g., “Industry 4.0” AND “product design”) I am not convinced that the keyword is effective. Probably, many important papers do not use the term Industry 4.0 and sensor simultaneously in the most critical parts of the abstract, title, or keywords because IoT/sensors is a pillar of industry 4.0.

Section 3

The sentence “As evident from Table 3, the significant majority of articles on industry 4.0, sensors 141 and product design rank on the Q1 best quartile index” is not accurate. Most of the articles in the authors' sample belong to lecture notes.

I do not see the relevance of Table 3. Too long and does not provide specific information to the reader. I suggest presenting the thematic areas with a treemap

Please improve the quality of figure 1 and 2 (perhaps expanding the figure and removing the borders)

Please check the dates, October 2022 or May 2022?

Figure 3 is unreadable. VOSViewer allows making adjustments to the network so that you can increase the attraction of the node on the right to make the figure more legible. Figure 4 and 5 are also impossible to read. Bibliometric networks are important to understand clusters and keywords in each cluster and then discussing it deeply. Merely presenting the networks is not a bibliometric analysis. It is necessary to explain and discuss the topics, assisting the reader in understanding the relationships.

“are presented in Figure 4, making clear the net-212 work of keywords that appear together/linked in each scientific article”: no, it is not clear. The authors need to clarify what is included in the network.

Section 4 

This part seems to be the systematic review of the 35 selected papers (should be clearly stated). I suggest that authors include the table of references with a summary of each one, for example, the type of study (conceptual/empirical, the approach, main conclusions), and then make the analysis.

Section 4.1. is strange. The reader is expecting to understand the details of product design. There are many other important studies only discussing Industry 4.0, the sample is not appropriate for this purpose. For example, the second paragraph presents the origins of the term Industry 4.0; this is not important on page 8 of the paper.

Section 4.1.1. is also addressing vague aspects. Are the authors restricting the discussion to product design? It should be made more clear. Figure 6 should be revised to provide an overview of Industry 4.0 technologies for product design.

Section 4.2 is more interesting because it is the part of Industry 4.0 where the authors aim to contribute (not big data, AI, or others earlier discussed that only confuse the message). However, the paragraphs do not seem specific to product design; improving the focus on product design is necessary to make an innovative contribution. The reader is not expecting sentences explaining what a sensor is; this section needs details.

The remaining sections after 4.2 are very interesting, and now the reader can find a contribution to the field. In my opinion, the previous part needs a significant change, or the authors will lose the reader at the beginning of section 4. I suggest separating the Industry 4.0 vision in product design from the interesting contributions that the authors provide in sections 4.3-4.5.

I suggest including a section with an agenda for future research before the conclusions or as a subsection of the conclusions.

Section 5

Please remove vague sentences about Industry 4.0; this is not the place to restate what Industry 4.0 is (this is a very well-known concept). The authors should conclude with specific statements for product design.

Study limitations are missing.

In summary, the topic (specific to sensors and product design) is extremely interesting. The authors present a good analysis in sections 4.3 – 4.5. However, the rest of the paper needs many improvements, particularly because the bibliometric analysis is absent, and there are many vague statements about industry 4.0 (that can appear in the early stages of the paper, but after that, the reader needs details for product design). I hope my comments can assist the authors in their work's next steps to improve product design knowledge in Industry 4.0.

Author Response

The authors are grateful for their availability and suggestions to improve the article,
thanks

Reviewer 2 Report

The paper's topic is interesting and is in the range of journal topics. The article provides a detailed discussion of the issues surrounding the opportunities and challenges that exist in the ongoing transition towards smart factories and products. The paper is based on a systematic bibliometric literature review (LRSB) methodology to explore and synthesize data on how Industry 4.0 and sensors can leverage product design. 

In my opinion, the novelty of this paper is not presented. The discussion presented in section 4 should be more supported by various publications. The authors should emphasize their own observations related to the main goal of the work. It may also be of interest to discuss further research in the conclusion section.

Author Response

(The authors gave the same response as above.)

Reviewer 3 Report

accepted

Author Response

The authors are grateful for their willingness to review the article,
thanks

Round 2

Reviewer 1 Report

Dear authors, thank you for the opportunity to read your paper. Many improvements were made to the previous revision. The following parts are not yet entirely clear:

4. The paragraph “The procedure started with using the keyword” is not clear. It seems that the authors screened the titles, abstracts, and keywords of 23,300 references, which seems highly improbable. Not even the 2,870 records seem feasible to screen.

The authors seem to be explaining the keyword testing and experiments before selecting the better focus of the analysis, not the screening process. I suggest including a figure of the process or improving the presentation because it is not clear how the 35 papers appear (seems to be the result of TITLE-ABS-KEY ("Industry 4.0" + AND sensors AND + "Product Design" but it is not stated clearly). 

Answer: Thanks to the reviewer for pointing out the sentence construction and opportunity for improvement. Indeed, the way the methodology is described can create confusion in the reader. Thus, we revised the entire paragraph in order to make the methodological approach clearer

New Comment: Thank you for the answer, the text was improved. However, it is confusing when the authors state that " only articles in journals considered of "high quality" should be synthesized in a literature review" but the table includes conference papers.

__

5. To increase the confusion, the authors state that “only articles in journals considered as "high quality" should be synthesized in a literature review” but most of the papers are presented at conferences.

Answer: We understand the reviewer's concern and appreciate the opportunity to clarify this point. The papers presented at conferences were published in the respective proceedings, which are indexed in Scopus. For this reason, we also consider these papers to be of high quality.

New Comment: I understand the interest in including conferences, but the decision contradicts the previous sentence about the selection of journals only - the methodological reference selected by the authors. The solution is to change that part extracted from the methodology because the authors do not restrict to journal

_____

7. After madding some tests (e.g., “Industry 4.0” AND “product design”) I am not convinced that the keyword is effective. Probably, many important papers do not use the term Industry 4.0 and sensor simultaneously in the most critical parts of the abstract, title, or keywords because IoT/sensors is a pillar of industry 4.0.

Answer: We understand the reviewer's point of view. However, regarding the object of study under the title “How Industry 4.0 and Sensors Can Leverage Product Design: Opportunities and Challenges”, we consider that these keywords are the fundamental ones. For future studies we will certainly research other methodological processes.

New Comment: In this case, this study's limitation must be stated near the other limitations in the conclusions.

____

1. The sentence “As evident from Table 3, the significant majority of articles on industry 4.0, sensors and product design rank on the Q1 best quartile index” is not accurate. Most of the articles in the authors' sample belong to lecture notes.

Answer: We understand your concerns and appreciate the opportunity to improve. Therefore, we eliminated the lecture notes from our systematic literature review with Bibliometric Analysis to raise the quality of the research.

New Comment: It still is not consistent because most of the papers do not have quartile so it is not possible to state that the majority ranks Q1 (I do not see a problem in including Lecture Notes, it is the authors' decisions, the problem is the sentences about the sample that must be coherent).

____

6. “are presented in Figure 4, making clear the network of keywords that appear together/linked in each scientific article”: no, it is not clear. The authors need to clarify what is included in the network.

Answer: Thank you for reading our manuscript carefully and providing very constructive comments. This is certainly an important concern. We carefully revised the bibliometric analysis and explained and discussed the topics.

New Comment: It was not answered. For example, each color is a cluster - which are those clusters, and what do they represent?

______

5. I suggest including a section with an agenda for future research before the conclusions or as a subsection of the conclusions.

Answer: This is clearly an important question. We strongly agree with the reviewer and, therefore, added some suggestions to take into account for future studies.

New Comment: Please clarify the answer and the location of future work agenda, I could not identify the new suggestions.

Author Response

Dear,
The suggestions were contemplated and were precious in improving the article,
Thanks,

Reviewer 2 Report

The authors have improved the paper. They have taken into account some suggested suggestions mentioned in the review. The article is written clearly and is well structured by its aim.

Author Response

Dear,
The suggestions were precious in improving the article,
Thanks,